# Probabilistic Forecasting with Coherent Aggregation

## Abstract

Obtaining accurate probabilistic forecasts while respecting hierarchical information is an important operational challenge in many applications, perhaps most obviously in energy management, supply chain planning, and resource allocation. The basic challenge, especially for multivariate forecasting, is that forecasts are often required to be coherent with respect to the hierarchical structure. In this paper, we propose a new model which leverages a factor model structure to produce coherent forecasts by construction. This is a consequence of a simple (exchangeability) observation: permuting base-level series in the hierarchy does not change their aggregates. Our model uses a convolutional neural network to produce parameters for the factors, their loadings and base-level distributions; it produces samples which can be differentiated with respect to the model's parameters; and it can therefore optimize for any sample-based loss function, including the Continuous Ranked Probability Score and quantile losses. We can choose arbitrary continuous distributions for the factor and the base-level distributions. We compare our method to two previous methods which can be optimized end-to-end, while enforcing coherent aggregation. Our model achieves significant improvements: between $11.8 - 41.4\%$ on three hierarchical forecasting datasets. We also analyze the influence of parameters in our model with respect to base-level distribution and number of factors.

## 1 Introduction

Obtaining accurate forecasts is an important step for long-term planning in complex and uncertain environments, with applications ranging from energy management to supply chains, from transportation to climate prediction Hong et al. (2014); Gneiting & Katzfuss (2014); Makridakis et al. (2022). Going beyond point forecasts such as means and medians, probabilistic forecasting provides a key tool for forecasting uncertain future events. This involves, e.g., forecasting that there is a 90% chance of rain on a certain day, or that there is a 99% chance that people will want to buy fewer than 100 items at a certain store on a certain week. Providing more detailed predictions of this form permits finer uncertainty quantification. This in turn permits planners to prepare for different scenarios and to allocate resources depending on their anticipated likelihood and cost structure. This can lead to better resource allocation, improved decision making, and less waste.

In many applications, there exist natural hierarchies over quantities one wants to forecast: energy consumption at various temporal granularities, from monthly to weekly; or at different geographic granularities, from the building-level to city-level to state-level; or forecasting retail demand for specific items, as well as for categories of items or brands. Typically, most or all levels of the hierarchy are important: base-levels of the hierarchy are key for operational short-term planning; and higher levels of aggregation yield insights on longer-term or coarser trends. Moreover, it is often desired that the probabilistic forecasts at different granularities are coherent (or consistent) for efficient decision-making at all levels Hong et al. (2014); Jeon et al. (2019).

To be somewhat more precise, we say that probabilistic forecasts at different granularities in a hierarchy are *coherent* if and only if there exists a valid joint distribution across all base-levels such that the probabilistic forecasts have the same distributions as the corresponding marginals of the joint distribution Rangapuram et al. (2021); Jeon et al. (2019); Taieb et al. (2017). See also Def. 2.1 below. This follows Taieb et al. (2017), and informally it means that the distribution of an aggregate is the sum of the distributions of the

| Method | Coherence | Differentiable samples | Arbitrary loss function | Factor model | Arbitrary factor/ base distribution | #Parameters to represent a forecast |
|---|---|---|---|---|---|---|
| Deep HierE2E | ✓ | ✓ | ✓ | ✗ | ✗ | Low |
| DPMN | ✓ | ✗ | ✗ | ✓ | ✗ | High |
| This work | ✓ | ✓ | ✓ | ✓ | ✓ | Low |

Table 1: Desirable properties satisfied by the models Deep HierE2E (Rangapuram et al., 2021; Olivares et al., 2023) and DPMN (Kamarthi et al., 2022) and the proposed method. The ideal method is 1) coherent by construction, 2) differentiable with respect to its parameters for efficient optimization of expected loss functions 3) capable of optimizing arbitrary sample-based loss functions, 4) hierarchical in structure, represented by a factor model, 5) flexible in the choice of factor and base-level distributions, and 6) able to produce compact and expressive forecasts for ease of storing predictions. Our proposed method satisfies all of these desired properties.

base-levels in the aggregate. Designing a model which is accurate at all levels of aggregation of the hierarchy, and which can exploit information at different levels, while also enforcing coherency, is well-known to be a difficult challenge Hyndman et al. (2011). In particular, one can *not* simply aggregate or disaggregate probabilistic forecasts independently (assuming, of course, that one wants to achieve reasonable accuracy), without accounting for correlations among base time series.

In the last few years, end-to-end trainable neural network models have achieved a measure of success for (multi-horizon) probabilistic forecasting for univariate time series Wen et al. (2017); Flunkert et al. (2017); Alexandrov et al. (2020); Eisenach et al. (2020); Benidis et al. (2022). Compared to previous auto-regressive methods, these models provide additional flexibility: one can now fit quantile functions directly through nonlinear quantile regression Flunkert et al. (2017); Wen et al. (2017); Wang et al. (2019); Eisenach et al. (2020); Kan et al. (2022); Benidis et al. (2022) (while forbidding quantile crossing Park et al. (2022)); and one can differentiate through sampling complex hierarchical graphical models. The added flexibility results in higher forecast accuracy.

In addition to multi-horizon univariate time series forecasting problems, targets to be forecasted sometimes lie in a linear subspace of a common multivariate target. This is the case for hierarchical forecasting: there is a linear relationship between base-level series and aggregates Taieb et al. (2017); Jeon et al. (2019). Despite the flexibility of neural network models, we cannot expect the output of these models to learn from the training data to satisfy (these hierarchical, or other) constraints exactly Amos & Kolter (2017); Négiar et al. (2023). Recent work has aimed at enforcing these constraints exactly; and these neural networks models have achieved state of the art results for the *hierarchical* forecasting setup Rangapuram et al. (2021); Olivares et al. (2023); Kamarthi et al. (2022). These end-to-end forecasting models mitigate an important shortcoming of previous (pre-neural network) hierarchical forecasting methods: the need to forecast first, before reconciliating those forecasts in a coherent manner. By exploiting an end-to-end training approach, these methods permit one to train a coherent model in one step: either by integrating the reconciliation as a differentiable module Rangapuram et al. (2021); or by designing a probabilistic model which enforces the coherence, with distribution parameters given by neural networks Olivares et al. (2023).

In the light of the recent literature, here are properties which a hierarchical forecasting method should satisfy: 1) coherence by construction; 2) end-to-end trainability; 3) ability to optimize for arbitrary sample-based loss functions, e.g., quantile loss, Continuous Ranked Probability Score (CRPS), depending on the use-case; 4) exploitation of the hierarchical structure in the data, leading to a factor model representation; 5) flexibility in the choice of factor and base-level noise distributions that best approximate the data distribution; and 6) compact representation of the forecast to minimize storage cost.

In this paper, we present a method which satisfies all of these properties. We are aware of only two previous methods which provide end-to-end trainable, and coherent hierarchical forecasts: Rangapuram et al. (2021) and Olivares et al. (2023). However, these two previous methods only satisfy some of the properties stated above. See Table 1 for a summary.

In more detail, our **main contributions** are the following.

1. We propose a a novel model for probabilistic forecasting that satisfies all the desired properties stated above. Our model explicitly leverages exchangeability of the base-level targets using a factor model structure. It can be easily adapted to use different factor and base-level distributions, e.g., Gamma, Normal, Truncated Normal, etc. Our model enforces coherent aggregation exactly by construction.

2. The factor model parameters are the output of a neural network. We optimize this neural network directly by optimizing for marginal forecast accuracy, using a reparametrization trick. In addition, depending on the use-case, our model can be used for optimizing arbitrary forecast metrics, such as quantile losses, CRPS, mean squared error, or combinations of them.

3. Due to the importance of coherent forecasts in practice, we evaluate our method empirically by comparing against previous coherent end-to-end trainable methods, namely those of Rangapuram et al. (2021); Olivares et al. (2023); Kamarthi et al. (2022), on three public datasets. Following previous work, we evaluate on the CRPS metric Matheson & Winkler (1976), and we find that our method improves on previous methods by **11.8-41.4%** depending on the dataset. We additionally evaluate our mean forecasts using the relative MSE, and find that our method improves on previous methods by **28.9-44.1%** on two of three datasets. We analyze CRPS results at different levels of the hierarchy, demonstrating higher or comparable accuracy at all levels on the three datasets. Our model achieves this while providing a more compact forecast representation (an important practical consideration) than previous proposed coherent models.

4. We illustrate the influence of the choice of base-level distributions. Changing distributional assumptions, even in seemingly-minor ways, can have a large impact on accuracy, and the best choice often depends sensitively on the data. Our model has the flexibility to evaluate different modeling choices quickly and easily.

## 2 Background and Related Work

### 2.1 Probabilistic Forecasting

Probabilistic forecasts are usually formulated to quantify future uncertainty to inform decision making. We start by providing an overview of the general probabilistic forecasting problem Gneiting & Katzfuss (2014).

We focus on real-valued observations $y \in \mathbb{R}$, with $y \sim Y$, a realization of random variable $Y$. A forecast distributed as $\hat{Y}$ can be represented by an inverse cumulative density function (CDF) on the real line $\mathbb{R}$. At a given forecast creation date, we assume that we have access to a set of prior information, $\boldsymbol{X}$, which we use to inform our forecasts. This information could be historical observations of the variable we want to predict, static features about the entity, e.g., item characteristics in retail, correlated future features, e.g., future holidays which could influence the energy consumption we're forecasting, etc. We want to output the forecast $\hat{Y}$ which uses the most possible information from $\boldsymbol{X}$ to predict the distribution $Y$ from which $y$ is realized. Without access to the full distribution $Y$, we typically evaluate quality of $\hat{Y} \mid \boldsymbol{X}$ against a single realization $y$. If we estimate the conditional mean $\mathbb{E}[Y \mid \boldsymbol{X}]$, we set ourselves in the popular least squares regression setting Legendre (1805). If we estimate quantiles of $Y \mid \boldsymbol{X}$, we set ourselves in the quantile regression setting Koenker & Bassett (1978); Wen et al. (2017); Eisenach et al. (2020).

In practice, our forecasts are represented computationally using an inverse CDF, which depends on distribution parameters. These parameters are themselves the output of an (optimized) parametric function, e.g., a linear or generalized linear model or a neural network model. For example, if we constrain our forecast to be normally distributed, we can use its mean and standard deviation to characterize the full forecasted distribution. This mean and standard deviation will be the output of a parametric model, given covariates as input. More flexible models such as normalizing flows Rezende & Mohamed (2015) would require more parameters. Armed with a forecast class thus parametrized, we can optimize the fit of the distribution to the observation data. For this, we need a loss function which measures how well a proposed forecast in the forecast class fits the observations.

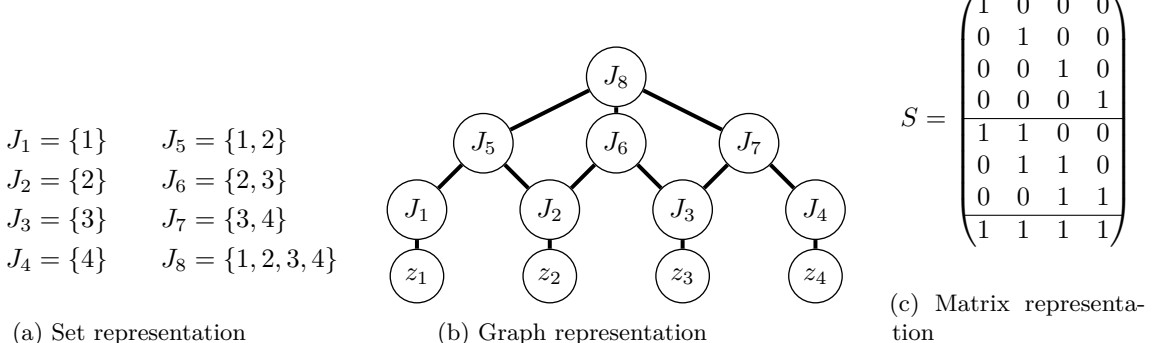

$$J_1 = \{1\} \qquad J_5 = \{1, 2\}$$
$$J_2 = \{2\} \qquad J_6 = \{2, 3\}$$
$$J_3 = \{3\} \qquad J_7 = \{3, 4\}$$
$$J_4 = \{4\} \qquad J_8 = \{1, 2, 3, 4\}$$

(a) Set representation

(b) Graph representation

(c) Matrix representation

Figure 1: **A simple hierarchical example of** $M = 8$ **aggregates over** $N = 4$ **entities.** Figure 1a (left) shows the set representation of the aggregation. Figure 1b (center) shows a possible graph representation: the edge between $J_6$ and $J_8$ could be removed since $J_6$ is included in the union of $J_5$ and $J_7$. The graph is a DAG, but it is not a tree: the node $J_2$ appears in the aggregations $J_5$ and $J_6$. Figure 1c (right) shows the corresponding aggregation matrix. In the matrix representation, we've added horizontal lines to separate *levels* of the hierarchy. These levels do not matter in our algorithm or methods, although they may be important for evaluation. These levels match the levels in the DAG representation, and they correspond to the topological ordering of the nodes in the graph. Our method uses only the matrix representation, which is equivalent to the set representation.

## 2.2 Forecasting with Coherent Hierarchical Aggregation

Hierarchical forecasting aims to solve the probabilistic forecasting problem, as discussed above, but over a set of variables with hierarchical relationships. Most often, we are interested in hierarchies where quantities are summed from one level to the level above it. For instance, the energy consumed in a region is the sum of the energy consumed in each zip code in the region. Methods concerned with hierarchies initially focused on mean forecasts Hyndman et al. (2011); Hyndman & Athanasopoulos (2013); Vitullo (2011); Hyndman et al. (2016); Dangerfield & Morris (1992); van Erven & Cugliari (2013); Wickramasuriya et al. (2019); Mishchenko et al. (2019). More recent methods consider probabilistic forecasts Olivares et al. (2023); Rangapuram et al. (2021); Han et al. (2021); Kamarthi et al. (2022). Mean forecasts are easy to aggregate: by linearity of the expectation operator, the mean of the aggregate is the aggregate of the means. However, this is not always the case for other quantities such as quantiles, including the medians.

There are two main method families for hierarchical forecasting: top-down disaggregation Vitullo (2011); Das et al. (2022); and bottom-up aggregation and reconciliation Hyndman et al. (2011); Hyndman & Athanasopoulos (2013); Taieb & Koo (2019). Recently, Das et al. (2022) have shown that, in a simplified setting, top-down disaggregation yields provably smaller excess risk than bottom-up aggregation. End-to-end coherent methods Rangapuram et al. (2021); Olivares et al. (2023) (discussed in more detail in Sec. 2.3 below) address the limitation in these two-stage approaches that information is inefficiently used (since models for each series are learned independently and are then post-processed to be coherent). These end-to-end methods fit all levels of the hierarchy simultaneously, with a module for reconciliating forecast at different levels, and they cannot be characterized as either bottom-up or top-down.

## 2.3 End-to-end Coherent Probabilistic Hierarchical Forecasting Models

We are only aware of two methods which yield provably coherent probabilistic forecasts and which allow the models to be trained in an end-to-end manner: Rangapuram et al. (2021) and Olivares et al. (2023). Two other works offer guaranteed coherent probabilistic forecasts Das et al. (2022); Taieb & Koo (2019), but they both rely on separately trained models, either for the base-level forecasts or for the top-level forecast. Wang et al. (2019) takes a similar approach to ours, motivated by exchangeability between time-series which share global

factors, but they do not consider the problem of aggregation. Here, we focus on the two end-to-end trainable coherent forecasting methods Rangapuram et al. (2021); Olivares et al. (2023). We compare properties of these models with the one developed in this paper in Table 1. The method described in Rangapuram et al. (2021) is designed for general convex constraints between multi-variate forecasts. Hierarchical forecasting is a special case of this general problem: it has added structure which is not leveraged by this method. On the other hand, the method described in Olivares et al. (2023) is too restrictive: it only considers mixtures of Poisson distributions. Since these distributions are integer-valued, this requires training the model using a likelihood loss which cannot directly be related to metrics closer to the downstream goal, such as CRPS or quantile loss on an important quantile level.

Finally, we should note that an important component of our approach to coherent aggregation is to make explicit use of the exchangability of information at the base levels of the hierarchy. To this end, Wang et al. (2019)'s observation that exchangeability of base time series induces a factor model structure is related. However, Wang et al. (2019) does not consider the hierarchical aggregation setting, and they do not consider end-to-end trainable models, but instead they suppose access to an already trained top-level model. From this perspective, we make broad changes to their method: 1) tailoring it to the coherent aggregation setting; 2) making it end-to-end differentiable; and 3) studying the influence of parametric distributions on accuracy.

## 2.4 End-to-end Probabilistic Hierarchical Forecasting Methods Without Coherence

In a related but separate thread, researchers have designed methods to regularize forecasts to be "more coherent" without enforcing coherence exactly. This is useful for datasets stemming from noisy measurements, e.g., disease control, where local and global measures are taken by different means to estimate quantities at different levels in the hierarchy. In this case, exact hierarchical aggregation does not hold. Han et al. (2021) focuses on regularizing quantile estimators for coherence between the different levels in the hierarchy. Recently, Kamarthi et al. (2022) proposed such a regularization approach for probabilistic forecasts, and focuses on the missing data case. Unlike methods described in the previous paragraph and our method, these methods do not provide guaranteed coherent forecasts between levels of the hierarchy (and thus we do not compare against them).

## 2.5 Notations

We consider the following hierarchical forecasting problem, with a set of base-level entities indexed by $[N] := \{1, \cdots, N\}$. Let $Z_1, \cdots, Z_N \in \mathbb{R}$ be the base-level quantities we want to forecast, and let $\boldsymbol{Z} = (Z_1, \cdots, Z_N)$. In addition, for $n \in [N]$, let $\boldsymbol{X}_n \in \mathbb{R}^D$ be random historical covariates observed for $Z_n$, and let $\boldsymbol{X} = (\boldsymbol{X}_1, \cdots, \boldsymbol{X}_N) \in \mathbb{R}^{DN}$ be all historical covariates across base time series. At each discrete time step $t \in [T]$, assume that new values of covariates $\boldsymbol{X}$ is given, and new value of targets are observed $\boldsymbol{Z}$. In addition, we introduce another set of indices $[U]$ to extend the framework into a contextual problem. We will refer to this dimension as the item dimension (e.g., in retail) or as the batch dimension. From an applied perspective, there may be multiple attributes or "dimensions" being used to describe the data, and we may or may not be interested in aggregating over all of them. For instance, if we are forecasting regional item-level demand in retail, then we may want to provide forecasts for many different items in the catalog at multiple regional granularities. The *item* dimension is different than the *region* dimension (over which we want to aggregate in this example). Hence, we reserve the subscript $u$ for the target variables and forecasts corresponding to such a dimension, which we do not use to aggregate across. At each time step $t$ and for each item $u \in [U]$, a forecaster is required to forecast for $\boldsymbol{Z}$, given realized values of $\boldsymbol{X}$. That is, let $z_{t,u,n}$ and $\boldsymbol{x}_{t,u,n}$ be historical observations for the quantity to be forecasted $Z_n$ and covariates $\boldsymbol{X}_n$ for each base-level entity $n \in [N]$. We consider $((z_{t,u,1}, \cdots, z_{t,u,N}), (\boldsymbol{x}_{t,u,1}, \cdots, \boldsymbol{x}_{t,u,N}))$ be a realized value of $(\boldsymbol{Z}, \boldsymbol{X})$.

We can model hierarchical aggregation in terms of a set of base-level entities $[N]$ and various subsets of that set. Suppose that we are interested in $M$ different hierarchical aggregations of the $N$ finest-grained entities. For each $m \in [M]$, we can define the set $J_m \subseteq [N]$ of fine-grained $Z$s in the $m$-th aggregate. We suppose that $\{J_1, \cdots, J_M\} \subseteq P([N])$ are subsets of the power set of $[N]$. The target variable corresponding to the $m$-th aggregate is $Y_m = \sum_{n \in J_m} Z_n$. For instance, if $N = 3$, we could have $J_1 = \{1, 2\}$ and $J_2 = \{2, 3\}$. The set

$\{Y_1, \ldots, Y_M\}$ is the set of aggregate targets in which we are interested. It could be that $Y_m$ correspond to a single base-level entity, in which case the corresponding $J_m$ is a singleton.

This setup allows us to define aggregations in matrix form. For $n \in [N]$, let $\boldsymbol{e}_n$ be the $n$-th canonical basis vector, i.e., the column vector with all zeros except for a 1 in the $n$-th coordinate. We define the vector $\boldsymbol{s}_m = \sum_{n \in J_m} \boldsymbol{e}_n$. For aggregate $m$, we have $Y_m = \boldsymbol{s}_m^\top \boldsymbol{Z}$. We therefore define the aggregation matrix

$$S = \left(\boldsymbol{s}_1 \cdots \boldsymbol{s}_M\right)^\top \in \{0,1\}^{M \times N}. \tag{1}$$

If there are $U$ different items, then $y_{t,u,m}$ is the $m$-th aggregated target variable corresponding to item $u$ at time $t$. The aggregation matrix $S$ does not depend on $u$ or $t$ in our work. Our method is very general. It applies to the case where we want to aggregate across the region dimension, or the time dimension, or both the region and time dimensions, or other dimensional of the data.

Regardless of the specific hierarchy, we want the probabilistic forecasts to be coherent with respect to the hierarchy. Here is an operational definition.

**Definition 2.1.** Let $\mathbf{Y} = (Y_1, \cdots, Y_M)^\top$ be a multi-variate random variable. Let $\overset{d}{=}$ denote equality in distribution. We say that $\mathbf{Y}$ is a *coherent aggregation* of the base-level series $\mathbf{Z}$ for the aggregation matrix $S$ if $\mathbf{Y}$ has the same distribution as $S\mathbf{Z}$, i.e., $\mathbf{Y} \overset{d}{=} S\mathbf{Z}$.

Note that historical observations $(y_{t,u,1}, \cdots, y_{t,u,M})$ are always coherent aggregations of base-level observations, as we can view historical observations as degenerate distributions with support at the observed values only.

Our aggregation matrix $S$ defines a directed acyclic graph (DAG), from the base-level entities up. Indeed, any DAG defines a possible hierarchy. In previous work Hyndman et al. (2011); Rangapuram et al. (2021); Olivares et al. (2023); Taieb & Koo (2019), the aggregation graph is assumed to be a tree, since latent variables are associated with each node, either to aggregate up or disaggregate down the forecasted quantities. In our method, it only matters which base-level series are in a given aggregate, i.e., only the aggregation matrix matters. Contrary to previous methods, we do not use relations between the aggregates (e.g., between $Y_m$ and $Y_{m'}$). Therefore, our method can handle aggregations represented by general DAGs, rather than just trees. In particular, in our method, the aggregates may overlap. See Figure 1 for an example, where we consider a hierarchy with $N = 4$ base-level entities and $M = 8$ aggregates. In this hierarchy, we are interested in the base-level entities themselves, in 3 different pairs of the base-level entities, and in the total across all entities. Some of the pairs have a non empty intersection.

## 3 Our Main Method

In this section, we present our main model for probabilistic forecasting with coherent aggregation.

### 3.1 Factor Model from Exchangeability

We develop a two-level probabilistic factor model for hierarchical forecasting that directly estimates the conditional joint probability function of base-level quantities $Z_1, \cdots, Z_N$ conditioning on historical covariates $\boldsymbol{X}$, i.e.,

$$p(Z_1, \cdots, Z_N \mid \boldsymbol{X}).$$

We can then obtain the target random variables in our hierarchical forecasting problems by marginalizing from this common joint distribution. A reasonable assumption in probabilistic forecasting applications is that the base level forecasts are exchangeable. Indeed, this was *implicitly* used in prior work Olivares et al. (2023). Recall that exchangeable random variables are those whose joint probability distribution does not change when the positions in a sequence in which finitely many of them appear are altered.

Under the assumption that $Z_1, \cdots, Z_N$ are exchangeable, conditioned on the covariates $\boldsymbol{X}$, then, motivated by de Finetti's theorem Diaconis (1977); Diaconis & Freedman (1980); Zaheer et al. (2017); Wang et al. (2019), we can model these exchangeable random variables as independent variables, conditioned on some

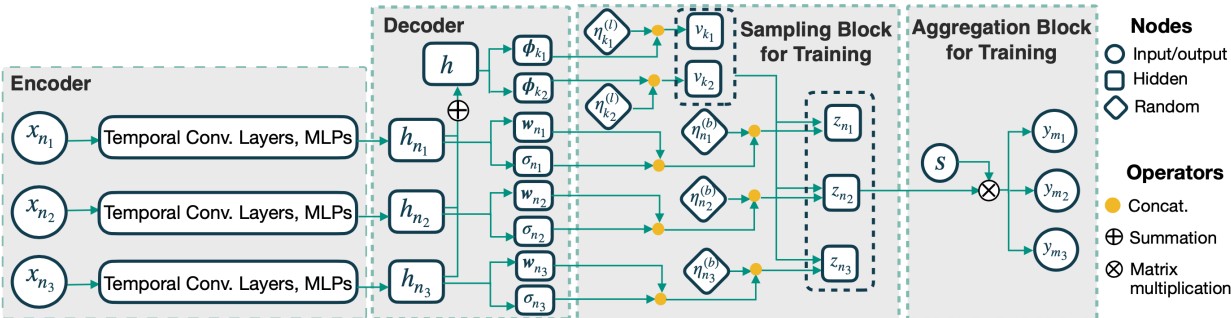

Figure 2: Model architecture for our work. We show an example with $N = 3$ base series, $M = 3$ aggregates, and $K = 2$ factors. Base-level series $\boldsymbol{x}_{n_1}, \cdots, \boldsymbol{x}_{n_3}$ are fed into a multi-variate neural network forecasting model such as MQCNN Wen et al. (2017). This model outputs encodings for each base-level series. These encodings are used in two manners: they are summed to produce an encoding at the common factor level, which is decoded into the factor distribution parameters $\boldsymbol{\phi}_{k_1}$ and $\boldsymbol{\phi}_{k_2}$ by a shallow network; they are also decoded directly by another shallow network to produce the base-level distribution loadings $\boldsymbol{w}s$ and parameters $\boldsymbol{\sigma}s$. The base-level forecast distributions can be sampled differentiably using a reparametrization trick by using parameter-free random inputs from factor level $\eta^{(l)}$'s and the base level $\eta^{(b)}$'s. Aggregating these samples $z_{n_1}, z_{n_2}, z_{n_3}$ with aggregation matrices $\boldsymbol{S}$ yields coherent samples at all levels of the hierarchy $y_{m_1}, \ldots, y_{m_3}$. Finally, aggregate samples are used to define the desired loss.

multivariate latent variables $\boldsymbol{v}$, such that

$$p(z_1, \cdots, z_N | \boldsymbol{X}) = \int p^{(l)}(\boldsymbol{v}|\boldsymbol{X}) \left[ \prod_{n=1}^{N} p^{(b)}(z_n | \boldsymbol{X}, \boldsymbol{v}) \right] d\boldsymbol{v}, \tag{2}$$

where $p^{(l)}(\cdot)$ and $p^{(b)}(\cdot)$ are, respectively, the probability functions of latent variable $\boldsymbol{v}$ and of the base-level quantity $z$, conditioned on the value of latent variable. Choosing the factor distributions and the base-level distributions are important design choices in our model. We will evaluate different choices empirically in section 4.3.

From Eqn. 2, we see that the target variables we want to predict are often influenced by common factors. For instance, if we are predicting precipitations in a country over different geographical granularities, then factors like topography may be important (mountainous regions will have different characteristics than coastal regions); and if we are predicting demand in a country, then information about the overall national demand may be important.

Since identifying such factors often requires expert knowledge, we aim to learn them. To do so, we consider the following generic factor structure, where there are $K$ independent random factors $V_k$ for $k \in [K]$, such that

$$p(\boldsymbol{v} \mid \boldsymbol{X}) = \prod_{k=1}^{K} p(v_k \mid \boldsymbol{X}). \tag{3}$$

Let each $V_k$ follow a $L^{(l)}$-parameter distribution $f^{(l)}$, with parameters $\boldsymbol{\phi}_k \in \mathbb{R}^{L^{(l)}}$, where $f^{(l)}(\boldsymbol{\phi}_k) \in \mathcal{F}^{(l)}$, the family of chosen distributions, e.g., Gamma, or Normal.

We suppose that the parameters of these factor distributions can be predicted using past observations of the target variables (at all levels of the hierarchy). In particular, let these factors each follow a $L^{(l)}$-parameter distribution, and we define $\boldsymbol{\phi}_k := \phi_k(\boldsymbol{X}; \boldsymbol{\theta})$, where $\boldsymbol{\theta}$ represents parameters of a neural network model, and $\phi_k : \mathbb{R}^{DN} \to \mathbb{R}^{L^{(l)}}$. The function $\phi_k$ maps covariates (representing historical, time-independent and known future features, e.g., day of week) to distribution parameters, e.g., the parameters of a continuous distribution, or parameters of a normalizing flow Rezende & Mohamed (2015). The dimension of the image of $\phi_k$ varies depending on the number of unique parameters required for determining the chosen distribution from the pre-specified distribution family.

Let $\boldsymbol{W} \in [0,1]^{N \times K}$ be a loading matrix learned with data, and let $\boldsymbol{W} = (\boldsymbol{w}_1^\top, \cdots, \boldsymbol{w}_N^\top)$, where $\forall\, n \in [N]$, $\boldsymbol{w}_n := w_n(\boldsymbol{X}_n; \boldsymbol{\theta})$ and $w_n : \mathbb{R}^N \to [0,1]^N$. Then each entry in $\boldsymbol{W}$ quantifies how much the target variable in base-level entity depends on the factors. We assume it can also be predicted from covariates. Conditioned on a realization $\boldsymbol{v} \sim \boldsymbol{V}$, by Eqn. 2, the target variables $Z_n$ for $n \in [N]$ are independent, and each follows a $L^{(b)}-$parameter distribution $f^{(b)}$ from a pre-specified distribution class $\mathcal{F}^{(b)}$. In addition, each distribution depends on parameters $\boldsymbol{\lambda}_n \in \mathbb{R}^{L^{n(b)}}$, where given $\boldsymbol{\sigma}_n := \sigma_n(\boldsymbol{X}_n; \boldsymbol{\theta})$ we let $\boldsymbol{\lambda}_n := \lambda_n(\boldsymbol{w}_n^\top \boldsymbol{v}, \boldsymbol{\sigma}_n)$. $\lambda_n : \mathbb{R}^{L^{(b)}} \to \mathbb{R}^{L^{(b)}}$, the mapping to base-level distribution parameters is assumed to be known, given $\boldsymbol{\sigma}_n$ and $\boldsymbol{w}_n^\top \boldsymbol{v}$; and $\sigma_n : \mathbb{R}^D \to \mathbb{R}^{L^{(b)}-1}$ is learned. Overall, we assume that base-level quantities follow the following two-stage factor model:

$$
\begin{aligned}
&\forall k \in [K],\ v_k \sim f^{(l)}(\boldsymbol{\phi}_k) \\
&\forall n \in [N],\ z_n \sim f^{(b)}\left(\boldsymbol{\lambda}_n \mid \boldsymbol{w}_n^\top \boldsymbol{v}, \boldsymbol{\sigma}_n\right).
\end{aligned}
\tag{4}
$$

With Eqn. 3 and Eqn. 4, the joint distribution of the $(Z_1, \cdots, Z_N)$ target variables in Eqn. 2 can be written as

$$
p(z_1, \cdots, z_N | \boldsymbol{X}) = \int \left( \prod_{k=1}^K p^{(l)}(v_k | \boldsymbol{\phi}_k) \right) \cdot \prod_{n=1}^N p^{(b)}(z_n | \boldsymbol{w}_n^\top \boldsymbol{v}, \boldsymbol{\sigma}_n)\, d\boldsymbol{v}.
\tag{5}
$$

Given the general model above, we can compute statistics related to the full joint distribution across multivariate targets in the hierarchy, or marginal distributions of aggregates along the hierarchy. Suppose we are interested in the marginal distribution of $Y_m$ with $J_m = \{n_1, n_2\}$, which contains $n_1, n_2 \in [N]$. Then, the probability function of $Y_m$ can be written as

$$
p(y_m) = \int_{z_{n_1} + z_{n_2} = y_m} p(z_1, \cdots, z_n)\, dz_{n_1} dz_{n_2}.
\tag{6}
$$

In practice, when the integral is not tractable, we can sample from the marginal distribution of the forecasted aggregate $\hat{Y}_m$ by sampling from the base-level series in $J_m$, and aggregating the samples. In the next sub-section, we discuss how to use these samples to optimize the model for different objective functions.

## 3.2  Sampling and Model Structure

**Differentiable sampling of the factor model.**    We design our method so that it can provide differentiable samples: we can differentiate samples with respect to our model's parameters. Recent work Figurnov et al. (2018); Ruiz et al. (2016); Jankowiak & Obermeyer (2018) has shown that one can sample in a differentiable manner from almost any continuous distribution. Our method exploits these results: if we can compute differentiable samples from the factor distributions and from the base-level distributions, we can compute differentiable samples for our forecasts, at any level of aggregation.

We now describe how to sample differentiably from the marginal distributions, following Kingma & Welling (2013); Figurnov et al. (2018). To do so, we write our samples as a function of 1) model parameters and 2) samples from a distribution which does not depend on the model's parameters. More formally, we can write the $k$-th factor realization, $V_k \mid \boldsymbol{\phi}_k$, and n-th base-level target realization, $Z_n \mid \lambda_n$, as

$$
\begin{aligned}
v_k &= \alpha^{(l)}(\boldsymbol{\phi}_k, \eta^{(l)}) \\
z_n &= \alpha^{(b)}(\boldsymbol{\lambda}_n, \eta^{(b)}),
\end{aligned}
\tag{7}
\tag{8}
$$

where $\eta^{(l)} \sim \tilde{f}^{(l)}$ is parameter-free noise (e.g., sampled uniformly from $[0,1]$, or from a standard Normal), and $\alpha^{(l)} : \mathbb{R}^{L^{(l)}+1} \to \mathbb{R}$.

To obtain samples of our forecasted distribution, we start by sampling the factor distributions. The parameters of the factor distributions come from a learned neural network. We use these factor samples as parameters for the base-level distributions. We can then compute samples from the base-level distributions, which are independent conditioned on the (shared) factor samples. Finally, we aggregate the base-level samples up to the desired $m$-th aggregate. The samples over all aggregates are coherent by construction.

**Structure of the model.** Expanding on the differentiable sampling module of Eqn. 7-8, we further explain the overall structure of the model. Let $R$ be total number of samples we want to produce during the training process. For time $t \in [T]$, item $u \in [U]$, $r \in [R]$, let the realized parameter-free noise $\boldsymbol{\eta}_{t,u,r}^{(l)} = (\eta_{t,u,r,1}^{(l)}, \cdots, \eta_{t,u,r,K}^{(l)})$, where all $\eta^{(l)}$s are i.i.d. sampled from $\tilde{f}^{(l)}$. Similarly, let $\boldsymbol{\eta}_{t,u,r}^{(b)} = (\eta_{t,u,r,1}^{(b)}, \cdots, \eta_{t,u,r,N}^{(b)})$, where all $\eta^{(b)}$s are i.i.d. sampled from $\tilde{f}^{(b)}$. Given covariates $\boldsymbol{X}_{t,u}$, we generate parameters for defining the predictive distribution by

$$\boldsymbol{\phi}_{t,u,k} = \phi_k(\boldsymbol{X}_{t,u}; \boldsymbol{\theta}), \qquad \forall\, k \in [K] \tag{9}$$

$$\boldsymbol{w}_{t,u,n} = w_n(\boldsymbol{X}_{t,u,n}; \boldsymbol{\theta}), \qquad \forall\, n \in [N] \tag{10}$$

$$\boldsymbol{\sigma}_{t,u,n} = \sigma_n(\boldsymbol{X}_{t,u,n}; \boldsymbol{\theta}), \qquad \forall\, n \in [N]. \tag{11}$$

Given parameters of the distribution, we further obtain a forecasted sample $\hat{y}_{t,u,r,m}$ for the target $y_{t,u,m}$ by

$$v_{t,u,r,k} = \alpha^{(l)}\left(\boldsymbol{\phi}_{t,u,k}, \eta_{t,u,r,k}\right), \qquad \forall k \in [K] \tag{12}$$

$$z_{t,u,r,n} = \alpha^{(b)}\left(\lambda_n(\boldsymbol{w}_{t,u,n}^{\top}\boldsymbol{v}_{t,u,r}, \boldsymbol{\sigma}_{t,u,n}), \boldsymbol{\eta}_{t,u,r}^{(b)}\right), \qquad \forall\, n \in [N] \tag{13}$$

$$\hat{y}_{t,u,r,m} = \boldsymbol{s}_m \boldsymbol{z}_{t,u,r}, \qquad \forall\, m \in [M]. \tag{14}$$

Recall that $\boldsymbol{\phi}$s are learned distribution parameters for random factors, $\boldsymbol{w}$s are loadings for these factors on each base-level series, and $\boldsymbol{\sigma}$s are additional parameters for defining distribution at the base level. The $v$s and $z$s are the factor-level and base-level predictions in samples; and the $y$s are the aggregate forecasts in samples. The last line, Eqn. 14, ensures that the aggregates are coherent *by construction.*

As shown visually in Figure 2, Eqn. 9-14 define the network structure at training time. In practice, we add some structure to the above-defined $\phi_k$ and $w_n$ functions: our network computes encodings $h_1, \ldots, h_n$ of each base-level series. These encodings are then used in two places: 1) we sum them to obtain a top-level encoding $h = \sum_{n=1}^{N} h_n$, which is then decoded by a shallow network into the factor distribution parameters $\boldsymbol{\phi}$; and 2) they are decoded by a separate shallow network into the loadings $\boldsymbol{w}$ and parameters $\boldsymbol{\sigma}$. At inference time, since the predictive distribution is defined by outputs from Eqn. 9-11, the sampling module Eqn. 12-14 can be discarded.

Differentiable sampling is implemented for many distributions of interest in several open source machine learning frameworks, e.g., PyTorch[1] Paszke et al. (2019), TensorFlow[2] Abadi et al. (2015) and JAX[3] Bradbury et al. (2018), making it extremely easy to implement for many different functional forms. We demonstrate this in the PyTorch code snippet in Figure 4 in Appendix B. We only need to change a single line of code to change our distribution assumptions at either level.

### 3.3 Optimizing the Model

Having differentiable samples allows us to optimize for *any* loss which is a differentiable function of forecasted samples. This could be losses on the marginal distributions, i.e., the aggregates, such as squared error loss, quantile losses for quantile levels of interest (0.5 if the median is important, 0.9 or 0.99 if the tails are important), or even weighted combinations of these losses. It could also be a function of the joint distribution, e.g., the energy score Gneiting & Raftery (2007).

It is common in the hierarchical forecasting literature to evaluate forecast performance on the marginal forecasts Taieb et al. (2017); Rangapuram et al. (2021); Olivares et al. (2023) using the Continuous Ranked Probability Score (CRPS) Matheson & Winkler (1976). We focus on this loss as an example, and we define it now.

**Definition 3.1.** Let $y_m$ be the target realized value of some underlying distribution $Y_m$. Let $\hat{Y}_m$ represent the forecasted distribution of $Y_m$, and let $\hat{y}_m$ and $\hat{y}'_m$ be independent samples of the forecast distribution. Then, the CRPS between the forecasted distribution and observed value, using samples from the forecast

---

[1] https://pytorch.org/docs/stable/distributions.html: see `rsample` methods.

[2] https://www.tensorflow.org/probability/api_docs/python/tfp/distributions

[3] See e.g. https://jax.readthedocs.io/en/latest/_autosummary/jax.random.gamma.html.

| Dataset | # Items ($U$) | Base ($N$) | Levels | Aggregated ($M$) | Time range | Frequency | Horizon (time steps) |
|---|---|---|---|---|---|---|---|
| `Tourism-Large` | 1 | 304 | 4/5 | 555 | 1998-2016 | Monthly | 12 |
| `Favorita` | 4036 | 54 | 4 | 93 | 1/2013 - 8/2017 | Daily | 34 |
| `Traffic` | 1 | 200 | 4 | 207 | 1/2008-3/2009 | Daily | 1 |

Table 2: Summary of publicly-available data used in our empirical evaluation. The `Tourism-Large` dataset Wickramasuriya et al. (2019) represents tourism visits to Australia between 1998 and 2016. The `Favorita` dataset Favorita et al. (2017) represents daily grocery sales in stores owned by the Favorita Corporación in Ecuador between 2013 and 2017. The `Traffic` dataset Taieb & Koo (2019) consists of daily occupancy rate for 200 selected car lanes in California Bay Area between 2008 and 2009.

distribution, is defined as $\mathbb{E}_{\hat{y},\hat{y}'\sim\hat{Y}_m}\ell_{crps}(\hat{y},\hat{y}',y_m)$, where

$$\ell_{crps}(\hat{y},\hat{y}',y_m) = |\hat{y}-y_m| - \frac{1}{2}|\hat{y}-\hat{y}'|. \tag{15}$$

The CRPS can also be written as the integral over all quantile levels of the corresponding quantile loss Matheson & Winkler (1976). We use the formulation above because it is an expectation: we can easily produce an unbiased estimator of the CRPS via Monte-Carlo sampling.

Let our model parameters $\boldsymbol{\theta}$ belong to a pre-determined parameter set $\boldsymbol{\Theta}$. Then, as we will be evaluating our model using the CRPS, we fit $\boldsymbol{\theta}$ by optimizing the following objective:

$$\min_{\boldsymbol{\theta}\in\boldsymbol{\Theta}} \frac{1}{TUR} \sum_{m=1}^{M}\sum_{t=1}^{T}\sum_{u=1}^{U}\sum_{r=1}^{R}\ell_{crps}(\hat{y}_{t,u,r,m},\hat{y}_{t,u,R+r,m},y_{t,u,m}), \tag{16}$$

where $\ell_{crps}$ is given by Eqn 15.

In practice, we use a neural network with the MQCNN architecture Wen et al. (2017) parametrized by $\boldsymbol{\theta}$ to output parameters of the distribution from the covariates $\boldsymbol{X}$. This architecture takes historical, static and future features which can be either numerical or categorical.

### 3.4 Discussion

Here, we compare our method with coherent probabilistic forecasting baselines. We consider the two coherent, end-to-end trainable methods in Rangapuram et al. (2021) and Olivares et al. (2023). For completeness, we also consider an ARIMA-based model with reconciliation implemented in Olivares et al. (2022); Garza et al. (2022).

From our perspective, the first method of Rangapuram et al. (2021) is "too general." It consists of a neural network model Flunkert et al. (2017) which produces probabilistic forecasts for all time-series in the hierarchy. This method is in fact more general than hierarchical forecasting, since it is designed to enforce any convex constraint satisfied by the forecasts; due to the constraining operation in the method, it has to revise the optimized forecasts. It does not leverage specifics of the hierarchical constraints, which are more structured than a general convex constraint. The model predicts parameters of Gaussian distributions for each time-series in the hierarchy, without coupling. Since the forecasts are not guaranteed to be hierarchically coherent, the model then couples samples from these Gaussian distributions by projecting them on the space of coherent probabilistic forecasts. Both the sampling operation Kingma & Welling (2013) and the projection are differentiable, allowing the method to be trained end-to-end. This model allows different modeling choices, although they are not explored in the initial paper, since Gaussians can be replaced by any distribution which can be sampled in a differentiable way, i.e., almost any continuous distribution Ruiz et al. (2016); Figurnov et al. (2018); Jankowiak & Obermeyer (2018). In Rangapuram et al. (2021), the projection operator ensures coherence, and correlations between base-levels are learned only by optimizing the neural network. In contrast, our proposed method produces forecasts for base-level series only, while relying on common factors

to encode correlations. This removes the need to forecast at all levels simultaneously, therefore reducing computational requirements if we are only interested in a subset of the aggregates.

On the other hand, the method described in Olivares et al. (2023) is "too restrictive." It can only handle learned mixture of Poisson distributions. It is in fact a special case of our model. If we suppose that there is a single non-parametric factor $V_1$ in our model that follows a multivariate discrete distribution with the supports and weights outputted by the neural network, and that conditioning on the realized value of the factor, the base-level distributions are Poisson (Eqn. 13), then we recover the model of Olivares et al. (2023). Our model represents a substantial generalization along both directions: we consider multiple independent factors and arbitrary distributions.

Observe that these two methods represent two extremes on the spectrum between non-parametric modeling and parametric modeling. In Rangapuram et al. (2021), we do not have access directly to the distributions of the marginals (i.e., the distribution of each aggregate time-series) since the model outputs samples from Gaussian distributions, and then it couples them with a projection. Due to this coupling, the forecasts follow an unknown, non-parametric distribution. On the other hand, in the Olivares et al. (2023) method, we can easily describe the marginals, since they are designed to follow a mixture of Poisson distributions. The same holds true for our proposed method.

Another difference between these two approaches is that the model of Rangapuram et al. (2021) optimizes the fit over the marginal time series of interest, under the coherence constraint, while in the model of Olivares et al. (2023), the training objective is likelihood-based, which in general does not directly optimize the evaluation metric of interest. In addition, their objective is fixed regardless of the set of marginal time series being evaluated. We design our method so that we can optimize marginal metrics of interest.

Our non-neural network-based baseline ARIMA-MinT-Boot consists of three steps. In the first step, we fit an auto ARIMA model (Hyndman & Khandakar, 2008) to each marginal time series. Then, we make the mean forecasts coherent by studying the covariance matrix of forecasted errors Wickramasuriya et al. (2019), using ordinary least squares. Lastly, to obtain probabilistic coherent forecasts, we apply a a bootstrap-based method Panagiotelis et al. (2023) on the coherent point forecasts. Although ARIMA-based methods do not show state-of-the-art performance for these datasets (Olivares et al., 2023), we include it for completeness as an example of a reconciliation method. Among approaches to reconcile point forecasts (such as Wickramasuriya et al. (2019); Taieb & Koo (2019), and approaches to extend them to probabilistic forecasts (Panagiotelis et al., 2023; Taieb et al., 2017), we only report results for ARIMA-MinT-Boot as they achieved the best CRPS results across most of hierarchical datasets studied in (Olivares et al., 2022).

## 4 Empirical Evaluation

In this section, we present our main empirical results. First, we describe the empirical set up. Second, we evaluate the proposed model, by comparing with two previous end-to-end trainable models, proposed in Rangapuram et al. (2021) and Olivares et al. (2023), as well as an ARIMA-based reconciliation model. Finally, we analyze our model's sensitivity to hyperparameters: 1) choice of base-level distribution; and 2) number of factors.

### 4.1 Setting

**Datasets.** In our analysis, we consider three qualitatively different (public) datasets: `Tourism-Large`, `Favorita`, and `Traffic`. They have different properties which are representative of more realistic non-public data, and forecasting all of them accurately requires substantial modeling flexibility. The `Tourism-Large` dataset represents the number of visitors to different regions in Australia. The goal is to forecast thousands of visitors, i.e., rescaling count data by 1000. The aggregation is done according to a hierarchy over region and purpose of travel, allowing us to test a case where the aggregate levels have overlap. The `Favorita` dataset is a large retail dataset, and it contains both count data (whole items) and real-valued data (items sold by weight) for over 4000 items. The aggregation hierarchy is regional. We use it to test our method on a (relatively) large-scale problem. Finally, the `Traffic` dataset contains sum-aggregates of highway occupancy rates. The initial rates are hourly, but (following Olivares et al. (2023)) the dataset we consider is daily, i.e.,

it uses rates already aggregated to the daily level for each highway bend as base-level series. The hierarchy in this dataset was defined randomly over highway bends. We use the same hierarchy as previous work. This allows us to test whether our model requires aggregations to be in line with correlation structures to achieve high accuracy. For all three datasets, the forecasted quantities are non-negative.

**Pre-processing and features.** For data preprocessing, we follow previous work Rangapuram et al. (2021); Olivares et al. (2023). Our models take in both numerical and categorical features for historical, static and future data, as allowed by the MQCNN architecture Wen et al. (2017). We describe these features in detail in Appendix A.1.

**Evaluation metrics.** Our main evaluation metric is a target-normalized CRPS Matheson & Winkler (1976). We compute the score described in Eqn. 15, and normalize it by dividing the result by the sum of all target values. We also evaluate mean forecasts by reporting ratio between mean squared error across forecasts in all levels over mean squared error of the naive forecast (which treats the previous observation in each time series as the point forecasts for all future horizons), which we call RelMSE.

**Hyperparameter search.** We consider limited sets of hyperparameters when tuning our model. Since all considered data are non-negative, we consider the Clipped Normal, Truncated Normal, Log-Normal and Gamma distributions as candidates for the base distribution. The Clipped Normal is a normal distribution where all the density at negative values are moved to being point mass at zero. The Truncated Normal on the other hand renormalizes the non-negative part of a normal distribution; there is no added weight on zero. We only consider Gamma random variables as factors, although we could choose any other continuous distribution. When reporting final accuracy results of our model on test set, we used the base distribution that performs the best in validation set, which is Clipped Normal in all three datasets. We determine the number of factors by using results from the validation set. For `Tourism-Large` and `Traffic`, we ran experiments for number of factors $K = \{1, 2, 4, 6, 8, 10, 15, 20, 30, 40\}$; for `Favorita`, due to memory constraints, we set number of factors $K = \{1, 2, 4, 6, 8, 10\}$. Likewise for the learning rate, we performed binary-search by hand on $[10^{-4}, 10^{-3}]$, and we chose the best learning rate according to results on the validation set. This light hyperparameter tuning shows that 1) our method is easy to optimize, and 2) we would get even better results by performing an automated hyperparameter search. Moreover, we only train our models on the CRPS, i.e., we do not yet test the effect of a discrepancy between train and test metrics. We leave this analysis for future work.

**Optimization** We fit each model using (stochastic) AdamW Loshchilov & Hutter (2017) (Adam Kingma & Ba (2014) with weight decay) using the $U$ dimension as our batch dimension. We use $10^{-5}$ weight-decay in all of our experiments, without tuning this parameter. `Tourism-Large` and `Traffic` both have a degenerate batch dimension, in the sense that $U = 1$, therefore we use full-batch gradients. For `Favorita`, batch size is 8.

## 4.2 Comparison with Previous Methods

We compare the proposed model to the DPMN model from Olivares et al. (2023), the HierE2E model from Rangapuram et al. (2021), and an ARIMA-based reconcilliation method (Wickramasuriya et al., 2019; Panagiotelis et al., 2023). Following previous work, we report the CRPS (normalized by the sum of target quantities) at all levels of the defined hierarchies; see Table 3 for the best model choices in our proposed family. For detailed definitions of all hierarchical levels of these datasets, see Appendix A.2. The ARIMA-MinT-Boot results are generated using (Olivares et al., 2022), with confidence interval computed based on 10 independent runs. Results for HierE2E is generated based on three independent runs using hyperparameters tuned by (Olivares et al., 2022). All metrics for DPMN are quoted from (Olivares et al., 2023) with identical experimental setting on all datasets.

Our model achieves lower overall CRPS for the test sets of all three datasets, improving on previous methods by 11.8%, 23.4% and 41.4% on `Tourism-Large`, `Favorita` and `Traffic`, respectively, as seen on the *Overall* rows of Table 3. For `Tourism-Large` and `Favorita`, our model achieves better accuracy at almost every single level of the defined hierarchy. On `Traffic`, our model achieves remarkably better results at the finer-grained level (level 3 and 4), but slightly worse accuracy at the higher levels of aggregation. It achieves strong

| Dataset | Metric | Level | Ours | DPMN-GroupBU | DPMN-NaiveBU | HierE2E | ARIMA-MinT-Boot |
|---------|--------|-------|------|--------------|--------------|---------|-----------------|
| Tourism -Large | CRPS | Overall | **0.1101 ± 0.0009** | 0.1249 ± 0.0020 | 0.1274 ± 0.0028 | 0.1456 ± 0.0061 | 0.1317 ± 0.0008 |
| | | Level 1 | 0.0349 ± 0.0028 | 0.0431 ± 0.0042 | 0.0514 ± 0.0030 | 0.0721 ± 0.0095 | **0.0277 ± 0.0011** |
| | | Level 2 | **0.0601 ± 0.0021** | 0.0637 ± 0.0032 | 0.0705 ± 0.0026 | 0.0943 ± 0.0059 | 0.0628 ± 0.0010 |
| | | Level 3 | **0.0959 ± 0.0027** | 0.1084 ± 0.0033 | 0.1068 ± 0.0019 | 0.1280 ± 0.0077 | 0.1150 ± 0.0008 |
| | | Level 4 | **0.1372 ± 0.0022** | 0.1554 ± 0.0025 | 0.1507 ± 0.0014 | 0.1682 ± 0.0083 | 0.1688 ± 0.0007 |
| | | Level 5 | **0.0552 ± 0.0016** | 0.0700 ± 0.0038 | 0.0907 ± 0.0061 | 0.0992 ± 0.0133 | 0.0750 ± 0.0014 |
| | | Level 6 | **0.1000 ± 0.0009** | 0.1070 ± 0.0023 | 0.1175 ± 0.0047 | 0.1361 ± 0.0069 | 0.1221 ± 0.0014 |
| | | Level 7 | **0.1652 ± 0.0008** | 0.1887 ± 0.0032 | 0.1836 ± 0.0038 | 0.1996 ± 0.0072 | 0.1976 ± 0.0010 |
| | | Level 8 | **0.2321 ± 0.0036** | 0.2629 ± 0.0034 | 0.2481 ± 0.0026 | 0.2670 ± 0.0073 | 0.2841 ± 0.0008 |
| | RelMSE | Overall | **0.0601 ± 0.0019** | 0.1113 ± 0.0158 | 0.2680 ± 0.0748 | 0.2058 ± 0.0443 | 0.1075 |
| Favorita | CRPS | Overall | **0.3080 ± 0.0201** | 0.4020 ± 0.0182 | 0.5301 ± 0.0120 | 0.5099 ± 0.1080 | 0.3968 ± 0.0007 |
| | | Country | **0.2116 ± 0.0117** | 0.2760 ± 0.0149 | 0.4166 ± 0.0195 | 0.3521 ± 0.1526 | 0.2652 ± 0.0005 |
| | | State | **0.2910 ± 0.0205** | 0.3865 ± 0.0207 | 0.5128 ± 0.0108 | 0.4860 ± 0.1120 | 0.3782 ± 0.0006 |
| | | City | **0.3095 ± 0.0205** | 0.4068 ± 0.0206 | 0.5317 ± 0.0115 | 0.5293 ± 0.1046 | 0.4028 ± 0.0008 |
| | | Store | **0.4201 ± 0.0162** | 0.5387 ± 0.0253 | 0.6594 ± 0.0150 | 0.6723 ± 0.0630 | 0.5410 ± 0.0010 |
| | RelMSE | Overall | **0.5381 ± 0.0368** | 0.7563 ± 0.0713 | 0.9533 ± 0.0201 | 1.082 ± 0.5473 | 1.1270 |
| Traffic | CRPS | Overall | **0.0181 ± 0.0028** | 0.0907 ± 0.0024 | 0.0704 ± 0.0014 | 0.0309 ± 0.0062 | 0.0731 ± 0.0050 |
| | | Level 1 | 0.0176 ± 0.0030 | 0.0397 ± 0.0044 | **0.0134 ± 0.0022** | 0.0157 ± 0.0327 | 0.0452 ± 0.0066 |
| | | Level 2 | 0.0176 ± 0.0030 | 0.0537 ± 0.0024 | 0.0289 ± 0.0017 | **0.0103 ± 0.0099** | 0.0470 ± 0.0057 |
| | | Level 3 | **0.0176 ± 0.0031** | 0.0538 ± 0.0022 | 0.0290 ± 0.0011 | 0.0160 ± 0.0093 | 0.0544 ± 0.0042 |
| | | Level 4 | **0.0195 ± 0.0028** | 0.2155 ± 0.0022 | 0.2101 ± 0.0008 | 0.0881 ± 0.0094 | 0.1459 ± 0.0047 |
| | RelMSE | Overall | 0.0232 ± 0.0536 | 0.1750 ± 0.0099 | 0.0168 ± 0.0026 | **0.0047 ± 0.0054** | 0.0624 |

Table 3: Results of our empirical evaluation. We report the CRPS score for each dataset (smaller is better) at various hierarchical levels (a level with lower number represents a more aggregated level, more details discussed in Appendix A.2). Average accuracy and its interval are computed based on three independent runs. Our model improves on previous methods on all datasets at all levels but Level 1 of `Traffic`. On `Tourism-Large`, our model improves on the previous state of the art by 11.8%. On the larger-scale `Favorita` dataset, our model improves by 23.4%. On `Traffic`, we improve the best result by 41.4% overall. Our model performs notably better at the finest granularity; our model's performance is stable across levels, whereas the other models perform better at the aggregate levels than at the base level. We evaluate our mean forecasts by computing the RelMSE score at the overall level (summing the total squared error at all levels, and normalize it by that of naive forecasts). Note ARIMA-MinT-Boot produces deterministic mean forecasts across model runs. Our model achieves lower RelMSE than other models on two datasets: 44.1% improvement on `Tourism-Large`, 28.9% on `Favorita`, but reaches higher RelMSE on `Traffic`, despite large gains in CRPS.

accuracy overall due to its ability to model the fine-grained series very accurately. In this dataset, contrary to the others, the aggregation matrix is defined somewhat randomly, by sampling base-level series. The base-level series within an aggregate therefore do not share special structure, unlike stores in a city for the `Favorita` dataset, or regions in a state in the `Tourism-Large` dataset. This lack of correlation structure may explain the slight performance lag experienced by our model at the higher levels of aggregation. Because RelMSE overweighs aggregated level mean forecast accuracy due to the nature of this metric, our method is associated with sub-optimal mean forecast accuracy overall.

## 4.3 Sensitivity of the Model to Design Choices

We analyze the sensitivity of the model to 1) the base-level distribution, and 2) to the number of chosen factors.

|                | Tourism-Large | Favorita | Traffic |
|----------------|---------------|----------|---------|
| Gamma | $0.1174 \pm 0.0044$ | $0.4817 \pm 0.2274$ | $0.0738 \pm 0.0629$ |
| Log-Normal | $0.2245 \pm 0.0905$ | $0.5268 \pm 0.1211$ | $1.0135 \pm 0.2967$ |
| Trunc-Normal | $0.1123 \pm 0.0038$ | $0.3922 \pm 0.0695$ | $0.0216 \pm 0.0033$ |
| Clipped Normal | $\mathbf{0.1101 \pm 0.0009}$ | $\mathbf{0.3080 \pm 0.0201}$ | $\mathbf{0.0181 \pm 0.0028}$ |

Table 4: Performance of our model for various choices of base distribution, where results are based on three independent runs. We provide overall normalized CRPS for factor models with Gamma distributed factors and various base distributions.

**Sensitivity to base-level distribution.** To evaluate how accuracy of the proposed model changes depending on the choice of base distribution, we train models for four different base distributions on each dataset, while using Gamma distributions to model the factors $V_k$. We report the CRPS results in Table 4. On all three datasets, Clipped Normal performs the best. On `Tourism-Large` and `Traffic`, Clipped Normal is better than other base-level distributions by a small margin. However, its accuracy is significantly better than others on `Favorita`, possibly due to the base-level series in the retail demand dataset is sparse, and a zero-inflated distribution is needed. For more details, see Appendix C. We also observed that Gamma base-level distribution performs worse than Truncated Normal in all three datasets. Finally, on two of the three datasets, the Log-Normal distribution performs poorly.

**Sensitivity to numbers of factors.** To obtain good results on a large scale dataset such as `Favorita`, Olivares et al. (2023) required the neural network to output parameters for up to 100 support positions and weights to approximate the empirical distribution of random factors times loadings. Our model allows for several factors, where each factor $v_k$ requires only two parameters, contained in $\phi_k(\boldsymbol{X}; \boldsymbol{\theta})$ (Eqn. 9, 12). Separating random factors and deterministic factor loadings rather than learning an empirical distribution approximation as in Olivares et al. (2023), our model can model more complex distributions while requiring fewer parameters. For example, we discovered that using one Gamma factor in our work already yields improved forecast accuracy for `Favorita`, compared to previous models. Moreover, we show how the choice of the number of factors impacts forecast accuracy, using an experiment on `Traffic` dataset as an example. We show the results in Figure 3 below. On this dataset, we observe that a single factor already performs better than previous methods. The best results are obtained with 20 factors.

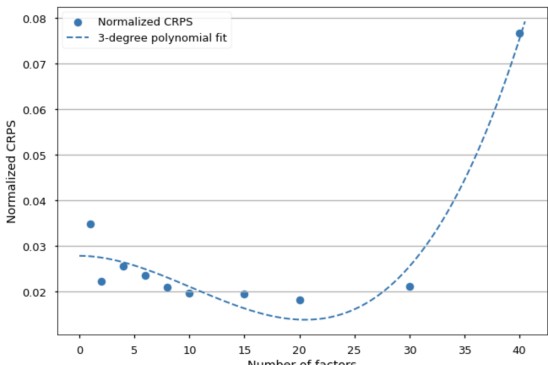

Figure 3: Performance of our model for different numbers of factors based on one run with the same random seed. We provide overall normalized CRPS on `Traffic` for a model with Gamma factors and Clipped Normal base distributions. We also fit a 3-degree polynomial function to the results.

## 5  Conclusion

In this work, we propose a novel probabilistic forecasting framework for hierarchical forecasting problems. To guarantee that probabilistic predictions are coherent in aggregation, our framework assumes that the predictive joint distribution over base-level targets follows a factor model, provided that base targets are exchangeable.

While the factor model assumption constrains the predictions, our model leverages recent advances in differentiable sampling, and it can optimize various sample-based objective functions that are aligned with forecasting evaluation metrics. We further conducted experiments on models in this framework on three benchmark datasets, comparing with alternative approaches. We demonstrate that our model improves overall forecast accuracy by 11.8-41.4% on three benchmark datasets. The model generates best or comparable forecast accuracy on almost all hierarchical levels for three datasets.

When forecasting for a given aggregate quantity within the hierarchy, our proposed model requires data for all base-level series included in the aggregate to be fitted on a single GPU. This may be prohibitive in certain applications where there exists thousands or millions of time series within an hierarchy. Removing this restriction would allow our method to handle to large-scale hierarchies. Our model currently uses a simple aggregation over all base-level embeddings for learning the parameters for the factors, this simple operator can lead to sub-optimal performance when forecasting at the most aggregated levels, as shown in empirical results. Leveraging other neural network structures with higher degrees of freedom for replacing the aggregation operator is a interesting future research direction.

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

# A  Details for each dataset

**Tourism-Large.**  The Tourism-Large dataset Wickramasuriya et al. (2019) represents visits to Australia, at a monthly frequency, between January 1998 and December 2016. We use 2015 for validation, and 2016 for testing, and all previous years for training. The dataset contains 228 monthly observations. For each month, we have the number of visits to each of Australia's 78 regions, which are aggregated to the zone, state and national level, and for each of four purposes of travel. These two dimensions of aggregation total $N = 304$ leaf entities (a region-purpose pair), with a total of $M = 555$ aggregates in the hierarchy. We pre-process the data to include static categorical features representing the region, zone, state and purpose. We use a time-varying categorical feature representing the month of year (an integer between 0 and 11), and finally, the past observed values of the time-series.

**Favorita.**  The Favorita dataset Favorita et al. (2017) contains grocery sales of the Ecuadorian Corporación Favorita in $N = 54$ stores. We perform geographical aggregation of the sales at the store, city, state and national levels, following Olivares et al. (2023). This yields a total of $M = 94$ aggregates. Concerning features, we use past unit sales and number of transactions as historical data. We include several static categorical features provided by the dataset, such as item type categories, or precomputed store clusters. Finally, to capture seasonality, we use day of week and day of month categorical features.

**Traffic.**  The Traffic dataset Taieb & Koo (2019) contains aggregates of daily freeway occupancy rates for 200 sampled (out of 963) car lanes in the San Francisco Bay Area between January 2008 to March 2009. We follow the aggregation defined in Taieb & Koo (2019). We note that this scheme aggregates occupancy rates by adding them up. There are three aggregated levels: four groups of 50 car lanes, two groups of 100 car lanes, and an overall group of 200 lanes. Each group was chosen randomly in Taieb & Koo (2019); we keep the same grouping. We follow previous experiments in the literature Taieb & Koo (2019); Rangapuram et al. (2021); Olivares et al. (2023), and split the dataset into training, validation, and test dataset of size 120, 120 and 126. In Table 3, we report accuracy numbers for the last date of 126 dates only, following the experimentation setting in (Rangapuram et al., 2021; Olivares et al., 2023).

## A.1  Features

**Tourism-Large.**  We describe the features in Table 5.

| Feature | Temporality | Kind |
|---|---|---|
| Observations | Past | Numerical |
| Month of year | Past/Future | Categorical |
| Region | Static | Categorical |
| Zone | Static | Categorical |
| State | Static | Categorical |

Table 5: Features used for the Tourism-Large dataset.

**Favorita.**  We describe the features in Table 6.

**Traffic.**  We describe the features in Table 7.

## A.2  Reported CRPS

For each dataset, CRPS is reported at different granularities, and then an overall CRPS is reported by taking a simple average across all levels. Below we discuss detailed definition of levels for each dataset, as shown in Table 3.

| Feature | Temporality | Kind |
|---|---|---|
| Observations | Past | Numerical |
| Store-level total transactions | Past | Numerical |
| Day of month | Past/Future | Categorical |
| Day of week | Past/Future | Categorical |
| On-promotion indicator | Past/Future | Categorical |
| Family | Static | Categorical |
| Class | Static | Categorical |
| Type | Static | Categorical |
| Cluster | Static | Categorical |
| Store number | Static | Categorical |
| City | Static | Categorical |
| State | Static | Categorical |

Table 6: Features used for the Favorita dataset.

| Feature | Temporality | Kind |
|---|---|---|
| Observations | Past | Numerical |
| Day of month | Past/Future | Categorical |
| Day of week | Past/Future | Categorical |

Table 7: Features used for the Traffic dataset.

**Tourism-Large** Base time series in `Tourism-Large` represents number of visitors from 4 purposes and 76 regions, where regions can be aggregated up to zones, states and nation. After counting aggregates at different levels, there are 555 time series. Level 8 consists of average CRPS across all $4 \times 76$ purpose-region level predictions. Level 7 consists of average CRPS across $4 \times 27$ purpose-zone level predictions. Level 5 and 6, each consists of $4 \times 7$ purpose-state level predictions, and 4 predictions for all purposes at the national level, respectively. Similar to levels 5-8, CRPS for different geographical granularities are reported at level 1-4, but the predictions being evaluated are for number of visitors aggregated across purposes. For example, level 4 includes CRPS averaged across 76 regions, and level 1 is CRPS for national level prediction.

**Favorita** For 4K grocery items sold across 54 stores across 22 cities in 16 states in Ecuador, the average CRPS across $4036 \times 54$ item-store level predictions are reported at the "store" level. At the "city" level, average metric across $4036 \times 22$ item-city level predictions are reported. Similarly, we measure forecast accuracy at the "state" and "country" level.

**Traffic** We follow definition of hierarchies for `Traffic` data in Taieb & Koo (2019). 200 sampled car lanes are randomly aggregated to four quadrants, and further two halves, and lastly into one group as a whole. Level 4 in Table 3 includes average CRPS across 200 car lanes. Level 3 and 2 each consists of average CRPS across four quadrants and two halves. Level 1 consists of CRPS for the aggregated prediction for all car lanes.

## B Code Script for Sampling

We provide a snippet of code for sampling from our factor model in Figure 4.

## C Visualization of Predictions

In Figure 5, we show our predictions for the `Favorita` dataset over a hierarchy containing Store 1, the city of Quito, the state of Pinchincha and the whole country of Ecuador.

```python
def get_samples(factor_params, shares, stds, aggregation_mat, n_samples):
    concentration = factor_params[..., 0]
    rate = factor_params[..., 1]
    # Change the following line to change factor distribution
    factor_samples = torch.distributions.Gamma(concentration, rate).rsample((n_samples,)
                                                                             )
    # Change the following line to change base distribution
    samples = torch.distributions.LogNormal((shares * factor_samples).sum(-1), stds).
                                             rsample()
    aggregate = Aggregate(axis=2, ndim=6)
    return aggregate(samples, agg_mat)
```

Figure 4: PyTorch function for sampling from our model, with a Gamma factor distribution and a Log-Normal base distribution. Note that the factor samples are shared across all base-level distributions. The samples are differentiable with regard to the function inputs. We can easily adapt this function to sample from other distributions. The parameters of the function are the outputs of a neural network.

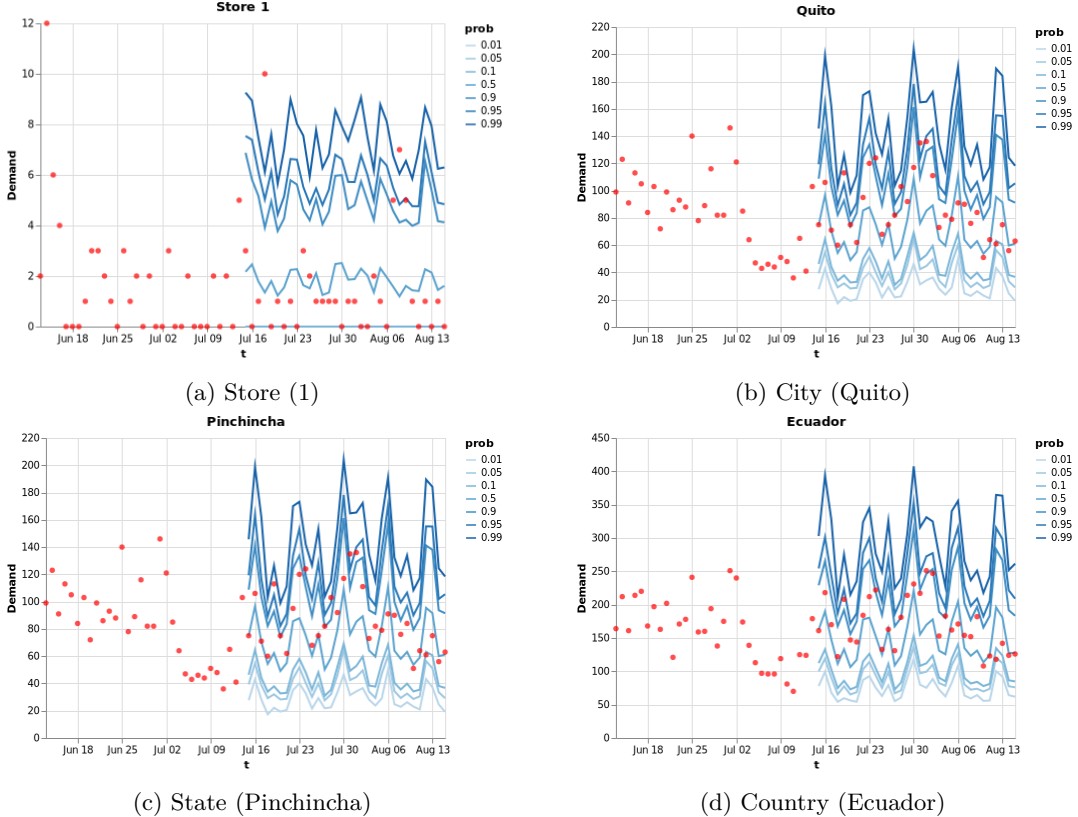

(a) Store (1)

(b) City (Quito)

(c) State (Pinchincha)

(d) Country (Ecuador)

Figure 5: Targets and predictions on the test set, for the hierarchy containing Store 1, for a given item, in the `Favorita` dataset. We visualize weekly forecast generated at the first forecast creation date in the test set. We show the forecasted quantiles at levels 0.01, 0.05, 0.1, 0.5, 0.9, 0.95 and 0.99 to demonstrate the spread of our forecasts, where the quantile forecasts are estimated empirically from 500 points from the factor model at each forecasted week. The model uses Gamma factors, and a Normal distribution clipped to be non-negative at the base-level. Clipping the Normal rather than truncating it allows to put point mass at zero, which is useful at the store level, as can be seen in Figure. 5a: up to P10 quantile forecast is zero at the store level for this item for all evaluation weeks.

