# OpenReview forum: "Probabilistic Forecasting with Coherent Aggregation"
_TMLR — Withdrawn by Authors_

### Review · Reviewer_Q4tv · 2023-08-25

**Summary Of Contributions:**

This paper introduces a model for probabilistic forecasting when coherent forecasts are required (the forecasts must be coherent given a hierarchical structure, eg certain variables sum up coherently). The model can handle different loss functions and distributions, and is fully differentiable (after using things like the reparameterisation trick). It can be viewed as a generalisation of the DPMN model of Olivares et al. (2023). The authors empirically test the model on 3 datasets, finding good performance compared to baselines, and look at how changing the base-level distribution and number of factors affects performance.

**Audience:**

Yes

**Claims And Evidence:**

Yes

**Requested Changes:**

Suggested change(s) (that I think would strengthen the work):
- The authors could move some plots from Figure 5 into the main text and discuss them more. Right now the results are purely a large table of numbers, and this could make it easier to understand.

Minor changes:
- It looks like there are wrong citations in the caption of Table 1. I think DPMN should be Olivares et al. (2023), and there should be no Kainarthi et al. (2022).
- Throughout the paper, the citation with/without parentheses is mixed up (use of \citet{} vs \citep{} in latex), and this should be fixed.
- At the top of page 5 (Sec 2.3), the authors say that 'hierarchical forecasting ... has added structure which is not leveraged by' Rangapuram et al. I don't think this by itself is a problem / bad thing. This issue more seems to be that Rangapuram et al. 'is designed to enforce any convex constraint satisfied by the forecasts' (as the authors say in page 10). Perhaps the authors could make this clear on page 5 as well.

**Strengths And Weaknesses:**

I thought the paper was very well-written and easy to follow. I appreciated the explanations at various points in the paper, and links to intuition. I view this is as a strong positive aspect of the paper. I also liked the (brief) discussion on limitations at the end of the paper.

In my opinion, the paper also provides sufficient evidence to support the claims made in the abstract and introduction. (The proposed model has the claimed properties, and the empirical performance is also as claimed.)

Given that I thought the paper well-written, I followed most of the logic, and also agreed with many of the arguments / choices made. Please see the requested changes for outstanding questions. I had one other questions about the writing/motivation:
- Despite the authors' explanations in the paper, I still do not understand why desideratum 4 on page 2 ('hierarchical structure leading to factor model') is so important. Is a factor model the only way to coherently model the hierarchical structure? Surely other ways may also be fine?

---

> ### Author Response · Authors · 2023-09-15
> **Response to Reviewer Q4tv**
>
> We thank the reviewer for their time, and their kind words.
>
> On the requested change: we agree with the reviewer that moving the plots to the main text will help with understanding the results. We will do so in a forthcoming revision.
>
> The goal of our factor model is to handle that the samples at the base-level are not independent, but correlated (sometimes highly so), due to unknown phenomena. We also want it to produce strictly coherent samples. A hierarchical factor model is not the only way to model strict coherence, as shown by the large forecast reconciliation literature that our paper discusses. On the other hand, having some kind of underlying factors helps for modeling correlations between the base-level variables. Our simple factor model is not the only way to model this; we could think of a hierarchical factor model (following the aggregation hierarchy), or using non-linear transformations of factors in the model.
>
> We will clarify the comment about Rangapuram et al, and fix the citations issues that the reviewer has underlined.

---

### Review · Reviewer_9suJ · 2023-08-28

**Summary Of Contributions:**

The paper introduces a novel hierarchical forecasting  (HTS) model that assumes a factor structure for the base-level series. Specifically, a CNN hypernetwork is employed to generate parameters for the factors, their loadings, and the base-level distributions.

The proposed model produces samples that are differentiable with respect to the model's parameters, offering the advantage of compatibility with a wide array of sample-based loss functions. Furthermore, the method is flexible in the sense that we can choose arbitrary continuous distributions for the factor and the base-level distributions.

Experiments conducted using three hierarchical forecasting datasets show that the proposed method provides better forecast accuracy compared to two end-to-end-trained baselines. Finally, the research considers the analysis of the impact of model parameters on the base-level distribution as well as the number of factors.

**Audience:**

Yes

**Broader Impact Concerns:**

Not applicable.

**Claims And Evidence:**

Yes

**Requested Changes:**

- Proposed model
	- Definiion 2.1 is not clear since there isn't an (explicit) predictive distribution or sample in the definition.
	- v is not defined in (2)
	- The authors make multiple assumptions without discussing them. I suggest clearly identifying these assumptions and practical design choices, and discussing their motivations.
		- "We suppose that the parameters of these ..."
		- "We assume it can also be predicted from coviariates"
		- "Overall, we assume that base-level quantities follow"
		- "In practice, we add some structure .. we sum them to obtain"
		- "In practice, we use a neural network with the MQCNN architecture"


- Given the empirical nature of the paper, simulated data should be considered to study the proposed model in a setting where the true underlying distributions are known.

- Baselines
	- Your choice of baselines seems arbitrary compared to existing literature. ARIMA-MinT-Boot h as not been used by recent papers on hierarchical probabilistic forecasting (Rangapuram et al. (2021) and Olivares et al. (2023). You should try to align your results with recent baselines used in the lierature, or at least add it in the appendix if the results are weaker.

- Datasets
	- It seems that previous work (Rangapuram et al. (2021) has used more than three datasets. Given the empirical nature of the paper, more datasets are needed to confirm the usefulness of the proposed model in hierarchical forecasting.


- "It is common in the hierarchical forecasting literature to evaluate forecast performance on the marginal forecasts"
	- Yes, but you also have research that has considered multivariate scores. See "Probabilistic forecast reconciliation: Properties, evaluation and score optimisation". This paper uses a sample-based loss function (energy score) which you could easily compute with your method.


- "It is in fact a special case of our model"
	- One paregraph is not enough. The authors should clearly show this formally (at least in the appendix)


- "In our method, it only matters which base-level series are in a given aggregate, i.e., only the aggregation matrix matters."
	- While trees have often been considered, the classical S matrix can model any aggregation of the bottom-level series (including overlaps). Hence, it is not clear how your approach is more general. In my opinion, your notations in Section 2.5 are equivalent to the definition of the S matrix. You are just explicitly defining the random variables for each aggregate series and the set of bottom-level series that are aggregated.


- "The added flexibility results in higher forecast accuracy.". This statement is a bit too strong.

- "Two other works offer guaranteed coherent probabilistic forecasts"
	- This paper does not consider probabilistic forecasting: "Regularized regression for hierarchical forecasting without unbiasedness conditions"


- The authors should clearly present the weaknesses of their approach. For exemple, one advantage of classical reconciliation methods is the fact that we can use very different univariate forecasting method for each series (including different distributional assumptions).

- Typo: "and new value of targets are observed Z.", "we consider ... be"


- "n-th base-level"
	- n, not in math mode.


- In (16), please define y_{t, u, u, r, m}

- Target-normalized CRPS is not defined

- Some notations and sentences are confusing
	- Section 2.1
		- Distributed as a random variable instead of a distribution.
		- "a realization of random variable Y" and later "to predict the distribution Y", "Without access to the full distribution Y"
	- Section 2.5
		- "base-level quantities", "random historical covariates"

	- "if we are predicting demand in a country, then information about the overall national demand may be important."
		- Did you mean for multiple cities of a country?

	- What do you mean by "joint probability function"? It is either PDF, CDF or probability distribution.

	- Page 9, what is an time? It should be defined early in the paper.

	-"we performed binary-search by hand". What does that mean?

	- "with confidence interval". Do you mean prediction interval?

**Strengths And Weaknesses:**

* Strengths
	* Hierarchical forecasting often involves a large number of time series with their aggregates. A factor structure is a natural assumption in this setting. Few research work has studied factor models for HTS.
	* Experiments show better forecast accuracy compared to baselines.
	* The sensitivity of the proposed model is analyzed with respect to the base-level distribution as well as the number of factors.

* Weaknesses
	* The paper essentially combines ideas from Wang (2019) and end-to-end hierarchical forecasting methods.
	* The authors did not consider simulated data to analyze the proposed model in a controlled environment.
	* In certain places, the paper is hard to read or confusing.
	* The assumptions and limitations of the proposed model are not clear compared to existing methods.

---

> ### Author Response · Authors · 2023-09-15
> **Response to Reviewer 9suJ**
>
> We thank the reviewer for their time. We appreciate the reviewer’s diligence and precision.
> We are working on a revised manuscript with the requested clarifications and precisions. In particular, we will expand on the weaknesses of our model.
>
> Note that we can choose different models/distributions for the base-level series or for the factors themselves and let our model select the relevant ones. We did not experiment with this for the sake of generality.
>
> - **Baselines.** The MinT Bootstrap is a recent method which is a stronger baseline than the ones considered in Olivares et al and Rangapuram et al. We will explain this, and add results for the more standard baselines, i.e. PermBU.
> - **Datasets.** We used the same three datasets as Olivares et al. In particular, we considered the large scale dataset Favorita which Rangapuram does not consider ; since this dataset is more realistic than those in Rangapuram et al (notably Tourism-small, Labor and Wiki), we did not want to distract the reader with numbers from other smaller and less realistic datasets.
> - **Synthetic data.** We will aim to include experiments on synthetic datasets which reflect the factor model structure we are interested in.
> Loss/evaluation: we will study the use of the energy score in both training and testing.

---

> > ### Comment · Reviewer_9suJ · 2023-09-28
> > **Response**
> >
> > Thank you for your response. Note that you did not address all my comments.
> >
> > I look forward to the revised version.

---

### Review · Reviewer_yfeM · 2023-08-31

**Summary Of Contributions:**

This manuscript looks at the challenge of making accurate predictions with uncertainty that have "coherent aggregation" within a hierarchy.  For example, getting distributions correct at both a local and regional scale.  They approach this challenge by using a factor modeling approach within a neural network predictor, allowing them to model correlations and relationships between samples.  This modeling approach seems to work well in pactice.

**Audience:**

Yes

**Broader Impact Concerns:**

None.

**Claims And Evidence:**

Yes

**Requested Changes:**

1) Add relationships to methods in spatial statistics
2) Add details on the reasonableness of the exchangeability assumption, including preferably evaluating it on the actual datasets
3) Evaluate differences in models to help identify the cause of the performance improvement.

**Strengths And Weaknesses:**

## Strengths

The method to construct factors within the framework appears novel and relatively straightforward to implement and evaluate.  Additionally, this framework is differentiable, so can be fairly easily constructed in modern frameworks.

The use of the CRPS metric is atypical and appears useful.

## Weaknesses

### Consideration of spatial statistics
The exchangeability assumption in Section 3.1 is not completely justified.  It may be a reasonable approximation, but in many of the motivating cases I don't think that it truly holds.  For example, the authors note
> "are predicting precipitations in a country over different geographical granularities, then factors like topography may be important"

This is true, but it ignores the spatial structure of the problem.  Typically you would also expect that neighboring locations are more similar than regions with similar topography but distant locations, which is inconsistent with the exchangeability assumption.  As such, the authors should spend more time addressing when this assumption is true and is a reasonable approximation.

Notably, there is also significant work on spatial quantile regression, including many probabilistic models, that do not use such a strong modeling assumption.  By modeling dependencies between regions they can straightforwardly get coherent aggregation.  There are obvious differences between these more classical approaches and the proposed work, notably the more flexible neural network approach, but this field should be considered and discussed in this work.

### Need to consider additional metrics

The authors only report the CRPS metric at the individual level, and RelMSE is only reported on the aggregate level. It is also common to provide coverage and other metrics such as log-likelihood.  Adding these common metrics and fully evaluating the data would give more credence to the method.

### Need additional ablation studies

It is unclear where the performance improvement comes from.  While there are some evaluations in Section 4.3, I am left wondering why this is doing better.  It would be great to elucidate the impact of the training loss specifically, which differs from prior work.  Are you simply doing better on CRPS because you are directly training on CRPS, whereas competing works are not?  It would be useful to show the sensitivity to that model choice.  In other words, is this doing better because the model is better, or because the loss is more relevant to the evaluation?

---

> ### Author Response · Authors · 2023-09-15
> **Response to Reviewer yfeM**
>
> We thank the reviewer for their detailed review. We answer their comments below.
>
> - **On spatial regression.** We thank the reviewer for pointing this field out. Our goal in the paper is to provide a general method for coherent aggregation: our method is applicable to aggregation across time, items (e.g. items in a category for retail application), geography and any other dimension of interest. For the special case of geographical aggregation, spatial methods are of great interest, and we will highlight this in a paragraph of related work in the revised paper.
> - **On our modeling assumption.** We will clarify this in our revised paper. Our initial assumption is that the aggregates are invariant to permutations in what is aggregated. We then simplified this by considering that the top-level aggregate is invariant to permutations in what is aggregated, and not considering all the lower-level aggregates in our factor model. An alternative would be to consider a hierarchical factor model, which would be harder to implement.
> - **Evaluation metrics:** We cannot compute the log-likelihood for our factor model in closed form, but only an upper bound. It does not make as much sense to report results for this upper bound, which is difficult to compare with log likelihood results from simpler distributions. CRPS is the standard metric in the literature because it is a proper scoring rule [1]. We can add coverage results for specific quantile levels. The Energy score is also a proper scoring rule, but previous literature has shown that it is not very sensitive to model misspecification [2,3].
> - **Concerning the loss:** for the same reason as above, we do not recommend training using negative log-likelihood: we only have access to an upper bound due to the factor model structure. We will try training with the energy score.
>
> [1] Tilmann Gneiting and Adrian E Raftery. Strictly proper scoring rules, prediction, and estimation.
> Journal of the American statistical Association, 102(477):359–378, 2007.
>
> [2] Pierre Pinson and Julija Tastu. Discrimination ability of the energy score. DTU Informatics,
> 2013.
>
> [3] Florian Ziel and Kevin Berk. Multivariate forecasting evaluation: On sensitive and strictly
> proper scoring rules. arXiv preprint arXiv:1910.07325, 2019.

---

> > ### Comment · Reviewer_yfeM · 2023-09-28
> > **I look forward to the revision**
> >
> > Hi,
> >
> > Thank you for these details.  I need to see the revised manuscript to fully evaluate these claims.  I look forward to seeing it.

---

### Author Response · Authors · 2023-09-15
**Overall response**

We thank the reviewers for their time, kind words and precise reviews. We understand that all three reviewers seem to consider our paper above the bar, while advising several changes which we agree will make the paper stronger, and better for readers.

We would like to take some time to make the following larger improvements, as suggested by the reviewers.

- **Ablation studies for the loss function at training time.** We will add results with models trained using the energy score. We won't consider the log likelihood since we can only compute an upper bound for our factor model, which makes it impossible to compare with other models.
- **Evaluation metrics.** We will add metrics for evaluation, in particular the energy score, coverage for certain quantile levels. For the same reason as above, we will not consider log-likelihood.
- **Synthetic data.** We will design a synthetic dataset with a factor model correlation structure to see what our model recovers, and test distribution mismatch.

We are also working on integrating all the precisions and clarifications requested by the reviewers, and will upload a revised draft when this is done.

We look forward to the continued discussion with the reviewers.

---

> ### Comment · Reviewer_9suJ · 2023-09-28
> **Response**
>
> Dear Authors,
>
> To avoid any misunderstanding, in its current form, i do not consider the paper "above the bar".

---

### Note · Authors · 2023-10-25

**Comment:**

Dear AE and reviewers,

It will take us longer to produce a revised version of the paper which answer the reviewers' most in depth concerns. We will therefore withdraw the paper for now, and resubmit later. We will make sure to choose the same AE when resubmitting. Thank you all for your input and work on this, as it is very valuable for improving the paper.

the Authors

**Withdrawal Confirmation:**

I have read and agree with the venue's withdrawal policy on behalf of myself and my co-authors.